# Gen-Review: A Large-scale Dataset of AI-Generated (and Human-written) Peer Reviews

## Abstract

How does the progressive embracement of Large Language Models (LLMs) affect scientific peer reviewing? This multifaceted question is fundamental to the effectiveness—as well as to the integrity—of the scientific process. Recent evidence suggests that LLMs may have already been tacitly used in peer reviewing, e.g., at the 2024 International Conference of Learning Representations (ICLR). Furthermore, some efforts have been undertaken in an attempt to explicitly integrate LLMs in peer reviewing by various editorial boards (including that of ICLR'25). To fully understand the utility and the implications of LLMs' deployment for scientific reviewing, a comprehensive relevant dataset is strongly desirable. Despite some previous research on this topic, such dataset has been lacking so far. We fill in this gap by presenting Gen-Review, the hitherto largest dataset containing LLM-written reviews. Our dataset includes 81K reviews generated for all submissions to the 2018–2025 editions of the ICLR by providing the LLM with three independent prompts: a negative, a positive, and a neutral one. Gen-Review is also linked to the respective papers and their original reviews, thereby enabling a broad range of investigations. To illustrate the value of Gen-Review, we explore a sample of intriguing research questions, namely: if LLMs exhibit bias in reviewing (they do); if LLM-written reviews can be automatically detected (so far, they can); if LLMs can rigorously follow reviewing instructions (not always) and whether LLM-provided ratings align with decisions on paper acceptance or rejection (holds true only for accepted papers). Gen-Review can be accessed at the following link: https://anonymous.4open.science/r/gen_review/.

## 1 Introduction

Since the release of ChatGPT in Q4 2022 (35), Large Language Models (LLMs) are revolutionizing many areas of our society (11). For instance, enormous potential for productivity growth has been reported in fields such as healthcare, software engineering, human-computer interaction, finance, and education, to name a few (21; 9; 30; 18; 8; 23; 49; 26; 47). From a broader perspective, LLMs are also expected to have a profound *impact on science in general*, regardless of their specific fields (6; 29).

LLMs can affect scientific work in various ways. They can be used to revise text (12), summarize prior literature (3), or implement an experimental pipeline or its parts (16). The use of LLMs for scientific work has initially faced ample criticism (2; 19; 31). However, LLMs are a valuable asset to researchers (6; 11) as they can facilitate routine scientific tasks, allowing researchers to focus on the scientific discovery. Consequently, efforts were made to promote a transparent disclosure of the usage of LLMs along the path leading to a scientific publication (1).

A complementary task, integral to the scientific process, is *peer-reviewing*. Some prior works have addressed the subject of using LLMs for peer-reviewing purposes, e.g., (28; 4; 25; 41; 46; 37; 24). As an almost anecdotal finding, the study of Liang et al. (28) reported that, after the release of ChatGPT, the reviews submitted to the 2024 edition of the International Conference of Learning Representations (ICLR) included a strikingly more frequent (up to 34 times) occurrence of words such as "meticulous" or "intricate", often associated with

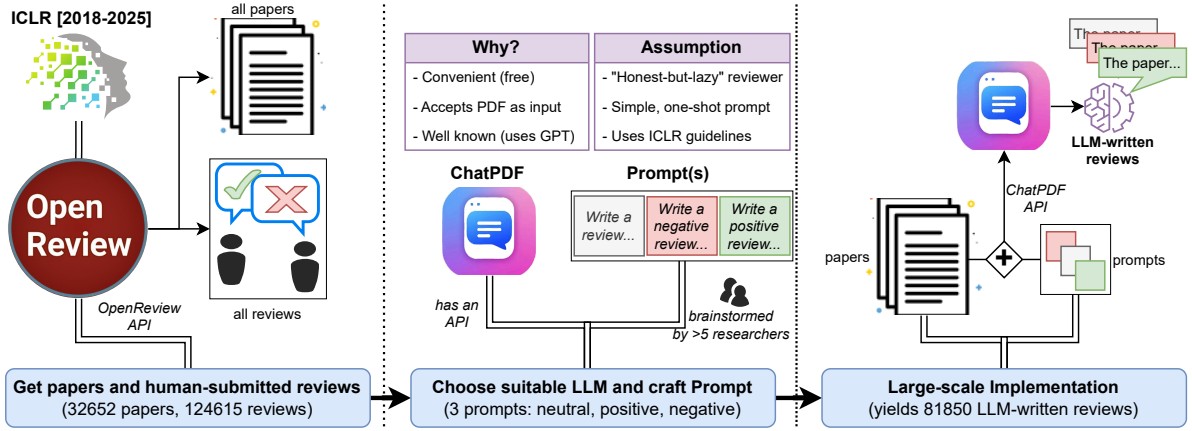

Figure 1: **The workflow to create `Gen-Review`.** We rely on the papers submitted to the [2018–2025] editions of ICLR (we also collect all of their human-submitted reviews). Then, we craft three simple prompts and we leverage the ChatPDF API to generate our large-scale dataset of LLM-written reviews. We then analyse our LLM-written reviews alongside those submitted by human reviewers.

ChatGPT, compared to the previous three ICLR conferences. Such an anomaly suggests that LLMs are likely being used for peer-review at top-tier conferences.

In fact, possibly as a response to the increasing number of papers that require peer-review, some established scientific outlets have started to actively integrate LLMs into their reviewing pipelines. For instance, ICLR'25 used LLMs to provide feedback to a subset of reviewers with suggestions for improving their reviews (50). As a result, 27% of reviewers confronted with such feedback updated their reviews (40). Yet, the overall sentiment towards a large-scale deployment of LLMs for reviewing remains mixed, with opinions ranging from "inevitable" to "a disaster" (32).

In light of such diverging opinions, it becomes apparent that the discourse on the impact of LLMs on scientific reviewing must be supported by fundamental data-driven research. To facilitate such research, we present `Gen-Review`, the hitherto largest publicly-available dataset of LLM-generated reviews. It contains over 80 thousand reviews generated for *all papers* submitted to the ICLR between 2018 and 2025. For each paper, three reviews were generated by issuing three independent prompts: one requesting a "positive" review, another requesting a "negative" review, and a "neutral" one without a specific instruction (our workflow is depicted in Fig. 1). We expect `Gen-Review` to foster investigations addressing LLM-driven reviewing, including but not limited to analyzing the potential bias in LLM reviews, gauging their overall quality, measuring the alignment of LLM-reviews with human-authored ones, and evaluating detectors of LLM-generated content. We illustrate the potential benefits of `Gen-Review` for such research by carrying out exemplary investigations. Specifically, after collecting all the human-submitted reviews for the same editions of the ICLR (which we provide in our dataset), we: *(i)* compare the LLM-proposed recommendation with the human-driven papers' outcome; *(ii)* investigate the presence of bias in our LLM-written reviews; and *(iii)* test a state-of-the-art detector of LLM-generated text, *Binoculars* (15), on our collected data.

**Contributions.** In summary, our paper makes two key contributions:

- We create `Gen-Review`, a large-scale dataset of over 80k LLM-written reviews, related to over 32k papers submitted to the [2018–2025] editions of the ICLR.

- We use our curated data to provide quantitative insights related to the utilization of LLMs for scientific peer-review.

This paper is organized as follows. First, we define our scope and justify the need for our contributions in Section §2. We describe the creation of `Gen-Review` in Section §3. Exploratory analyses are elucidated in Section §4. We discuss our results and provide avenues for future work in Section §5.

## 2 Preliminaries, Goals, and Motivation

We outline the context of our work, which also serves to substantiate some design choices (§2.1). Then, we outline our research goals (§2.2) and compare our contributions with related work (§2.3).

### 2.1 Background and Context

We summarize the landscape of using Artificial Intelligence (AI), such as LLM, for content generation. Then, we focus on the core of our work, emphasizing the relevance and necessity of similar efforts.

**Generative AI and LLMs.** One of the most appreciated capabilities of LLMs is their content-generation ability. An LLM can interpret the instructions embedded in a given *prompt* and produce a corresponding output. Initially, both the prompt and the corresponding output were limited to textual format (35). However, over time, LLM-related technologies substantially improved, and it is now possible to provide prompts (and requesting an output) as text, images, audio, videos, or a combination thereof (33). Recent findings have shown that the content generated by modern LLMs is of such a high quality that people can hardly figure out if it is human- or LLM-generated (13; 42; 7; 27).

**Detection of AI-generated content.** In some contexts (such as in science), determining the author of any given "creation" is of paramount importance (e.g., for authorship, or accountability). Therefore, due to the (allegedly) increasing appearance of LLM-generated content—such as in online social networks (27), or in emails (34)—there has been a growing interest in the development of *automated detectors* of LLM-generated media (39). Abundant prior works have developed various tools that can estimate whether a given input was generated by an AI (e.g., (22; 5)). For instance, Hans et al. (15) proposed *Binoculars*, an open-source detector that can infer whether a given piece of text was generated by, e.g., ChatGPT, with an accuracy of over 90% and a false-positive rate of only 0.01%. Unfortunately, attaining complete certainty on the true author of any given content is still an open problem: as stated in a recent survey (44), there is "an urgent need to strengthen detector research."

**LLM-assisted generation of scientific peer-reviews.** As acknowledged by the organizers/editors of various research venues (32; 50), *LLMs are being used today* in the peer-review of scientific articles. However, there are many ways in which LLMs can be used in this process (14). For instance, LLMs can take an existing review (or parts thereof) and improve its writing quality, or check that the review is written constructively and respectfully; LLMs can also provide a short and high-level account on a work referenced in a given submission; finally, LLMs can also write an entire review on the reviewers' behalf. Such a task can be carried out by *(i)* issuing a prompt such as "write a review on this paper" and *(ii)* attaching the PDF of the paper to review in the prompt. Doing so would produce an output text of variable length that describes the content of the paper and outlines its strengths and weaknesses—according to the LLM's judgment. For instance, a popular tool to achieve such an objective is ChatPDF:[1] by using its web interface (which is free), it is possible to produce a review of a paper in mere seconds (we provide a screenshot of ChatPDF's Web interface in Fig. 6).

**Concerns of AI-generated reviews.** Complete reliance on LLMs for reviewing duties raises various concerns, since the LLM's judgment replaces or influences that of the human expert. This can impact both the quality of the scientific selection of published works and the quality of the feedback returned to the authors. Among the most well-known issues of using LLMs for peer-review, we mention: the risk of "hallucinations" that undermine the correctness of the review; the lack of knowledge of the state of the art which prevents assessing the originality/novelty of the paper's claimed contributions; as well as the risk of breaching confidentiality agreements—due to uploading a submitted paper to a third-party. Consequently, certain venues have begun regulating the LLM usage for peer-reviewing purposes (e.g., NeurIPS'25) while others have explicitly prohibited any usage of LLMs in the reviewing process (e.g., CVPR'25). Regardless of whether LLMs are (or not) allowed, *what is crucial is being transparent towards the recipients of the reviews*: the authors have the right to be informed about whether LLMs played a role in the peer-review process of their papers (14).

---

[1] https://chatpdf.com/, allegedly the #1 PDF Chat AI; ChatPDF relies on the OpenAI GPT models.

## 2.2 Problem Statement and Research Workflow

At a high-level, our contributions are motivated by two complementary reasons: (i) the potentially inescapable integration of LLMs in (parts of) the peer-review process (32), which requires improving our generic understanding of LLM-generated reviews; and (ii) the necessity of identifying cases of misconduct wherein reviewers relied on LLMs without disclosure (thereby failing to uphold the authors' right to be informed (14)), which calls for ad-hoc detectors of LLM-generated reviews.[2]

Therefore, our first goal is the creation of a large-scale dataset of LLM-generated reviews, i.e., `Gen-Review`. We do this by using all paper submissions to the last eight editions of the ICLR. We elect to use ICLR papers as the core of the dataset and analysis not only because of their public reviews, but also because all ICLR submissions (including rejected or withdrawn papers) are publicly available. Crucially, this enabled us to create a dataset that is based on a large variety of papers in terms of quality (i.e., a dataset whose reviews are based solely on accepted papers would not be well-suited for research on the capabilities of LLMs in assisting in the peer-review).

Our workflow is depicted in Fig. 1 (further discussed in §3). Upon taking all the 32'652 papers submitted to the last eight editions of the ICLR (i.e., 2018–2025), we use ChatPDF to generate three reviews per paper, each based on an independent one-shot prompt: *(a)* a "positive" prompt, specifically crafted to induce the model to recommend an accept-class score; *(b)* a "negative" prompt, crafted to induce the model to recommend a reject-class score; and *(c)* a "neutral" prompt, wherein we do not add any explicit instruction on the (LLM-provided) recommendation. This led to the generation of 81'850 LLM-written reviews. Next, we collect all the human-submitted reviews (124'615 in total) for our sample of papers. Finally, we use all of this data to answer four research questions (RQ):

RQ1: *Is there any intrinsic bias in the LLM-written reviews?* (i.e., what is the general score distribution of "neutral" reviews w.r.t. "positive" and "negative" ones?)
RQ2: *How much do "neutral" reviews align with the overall outcome of the paper?* (e.g., if the LLM recommended accepting the paper, was the paper accepted?)
RQ3: *How much do LLMs fulfill the instructions provided in the prompt?* (e.g., if we specify a given length for the review, does the LLM follow such a requirement?)
RQ4: *How well can a state-of-practice detector (Binoculars (15)) identify the reviews in `Gen-Review`?* (and how does it perform on the human-submitted reviews?)

Given that ChatPDF relies on GPT-4o models, and that our papers are taken from ICLR, the answers to our RQs is restricted to this specific LLM and venue (both being, objectively, very popular).

## 2.3 Related Work

Various prior works have addressed problems related to our contributions. However, to the best of our knowledge, no existing dataset has a scope comparable to `Gen-Review`, and our findings are also original. In what follows, we summarize and compare the most related works to this paper.

**Lack of ground truth.** The findings of the seminal work by Liang et al. (28) indicate that LLMs are likely to have been used in ICLR'24. However, there is no ground truth to verify if any given review with an anomalous utilization of certain terms (e.g., "meticulous") was indeed written by an LLM. Moreover, without such ground truth, it is also impossible to determine the extent to which an LLM has been used (e.g., was it used to generate the entire review, or only to improve the textual quality of a human-written review?). The same shortcoming (i.e., lack of ground truth) also affects the work by Latona et al. (25), where GPTZero was used on the reviews submitted to ICLR'24, finding that potentially 15% were written with AI assistance. We address this problem by directly constructing a large-scale dataset of LLM-generated reviews, where the level and nature of AI involvement are fully controlled. Therefore, our dataset represents a valid proxy for a wide range of investigations.

**Small-scale analyses.** Thelwall et al. (41) assess ChatGPT's ability to predict the outcome of some papers submitted to ICLR'17 (collected in (17)); Vasu et al. (43) seek to find hidden biases in the reviews of LLMs,

---

[2]Ideally, such detectors can be used *before* the authors receive the LLM-generated reviews, so that action can be taken before making a (potentially inappropriate) decision on the paper's outcome.

but only consider 126 papers from ICLR'25. The authors of (37) compared the assessments of human reviewers to those of GPT-4 across 325 abstracts, finding alignment only for the best submissions. The datasets (and corresponding analyses) of both of these works are of a much smaller scale than ours. Our contributions seeks to provide a foundation for large-scale analyses.

**Limited-scope datasets of LLM-written reviews.** The closest works to our paper are those of Yu et al. (46), Kumar et al. (24), and the just-accepted NeurIPS'25 paper by Zhang et al. (48). All such works entailed the generation of LLM-written revriews based on submission to top-tier ML venues, but the datasets have a much smaller scope than our proposed `Gen-Review`. For instance, Yu et al. (46) generate the reviews by selectively removing some parts of the papers (such as the bibliography and images), and even though the reviews (16K in total; we have 81K) are based on papers submitted to the ICLR from 2021–2024, the overall number of papers used as a basis is only 500; Zhang et al. (48) carry out a broader analysis across 7.1K papers (mostly from ICLR'23–24), but our `Gen-Review` is much larger with 32K papers from *all* editions of ICLR since 2018. Kumar et al. (24) also use a much smaller number of papers (i.e., 1480 in total, taken from ICLR'22 and NeurIPS'22) and the reviews are generated by providing only the paper's text (i.e., without images) as input to the prompt. In contrast, our reviews are generated by providing the entire PDF, ensuring that the LLM has access to all the information available to any human reviewer.

**Orthogonal works.** There are also orthogonal works that propose datasets of various AI-generated content— not necessarily peer-reviews—such as (38; 10; 45); or works that focus on the detection of LLM-written *papers*—and not reviews—such as (31). Finally, we stress that our work is in no manner related to the detection of "fake reviews" in online platforms (e.g., online marketplaces (20; 36)).

## 3 `Gen-Review`: Large-scale Dataset of Peer Reviews

We describe the creation process of our major contribution: the `Gen-Review` dataset. Our workflow (shown in Fig. 1) can be split in three phases, which we elaborate on in the remainder of this section.

### 3.1 Preparation: retrieving papers and human-submitted reviews

We first outline the necessary requirements to reach our goal (see §2.2) and then explain how we collected the backbone of `Gen-Review`, motivating our decisions.

**Requirements.** To create a dataset of LLM-written peer-reviews, we need research papers—ideally (dozens of) thousands, since we aim to provide a dataset that enables large-scale assessments. Moreover, to provide a dataset that allows *fair* evaluations of LLM-written peer-reviews, we need papers that have been either "accepted" or "rejected": indeed, using only "accepted" papers would prevent one from gauging the quality of LLM-written reviews for those papers (theoretically of lower quality) that were not accepted to a given venue—which typically represent a large share of the submissions. Finally, we must ensure that our dataset includes also human-submitted reviews—which are necessary to facilitate comparison against LLM-written ones.

**Collection.** We determined that the ICLR is the most suited venue that fulfills all of the aforementioned requirements. Aside from being a top-tier venue, it yearly receives thousands of submissions; moreover, the complete peer-review details (including each human-submitted review, as well as outcome) of each submission are publicly observable—and there is historical data available on OpenReview for all of its editions. Therefore, we used the OpenReview API to collect all relevant data for our purposes for each paper submitted to ICLR from 2018 to 2025 (8 editions in total). In this way, we obtained: 32'652 papers (spanning accepted, rejected, and even withdrawn papers) and 124'615 human-submitted reviews (including their text, recommendation, and confidence). We do not consider submissions to satellite events of ICLR (e.g., workshops or blogposts). We note that such a process complies with OpenReview's terms of use (https://openreview.net/legal/terms).

### 3.2 Design choices: selecting the LLM, and crafting the prompts

The second step involves determining which LLM to use to generate our reviews, as well as devising prompts that would make `Gen-Review` appealing for future research. To better appreciate our contributions, we must

first describe our underlying assumption. Indeed, there are virtually infinite ways to craft a prompt that asks an LLM to "review a paper", and there are also dozens (or hundreds) of LLMs that can be leveraged for such a task. Therefore, to create `Gen-Review`, we set ourselves the goal to mimic a realistic and likely common use case. Specifically, we asked ourselves: "*If I were a reviewer tasked to write a review for a paper (submitted to ICLR) and I had no time to accomplish such a task, what would be the best way to do so by leveraging LLM-based solutions?*" Essentially, we assumed the perspective of an "honest-but-lazy" reviewer, who wants to fulfill their reviewing duties but does not have enough time to do so properly, and hence decides to rely on an LLM. This is a sensible assumption, given the increasing reviewing load in many research domains (32).[3]

**LLM-solution of choice: ChatPDF.** The first decision that our envisioned reviewer must make is which LLM to use. From this viewpoint, the ideal solution is one that fulfills the following criteria: *(i) it is convenient*—our reviewer does not want to spend money (e.g., to use more sophisticated models) or time (e.g., to setup a local model); *(ii) it is simple to use*—our reviewer just wants to write a prompt and provide the paper as-is, i.e., without converting the PDF into other formats; *(iii) it is well-known*—given that no LLM is intrinsically perfect, the reviewer (being a scientist) wants to resort to a solution for which there is evidence that it is "good enough" to carry out such a task. We found that ChatPDF is a solution that fulfills all of these criteria. Specifically, ChatPDF is free and is provided with a Web interface (even users who are not logged in can use it); it enables PDF upload by default[4], and it is popular, since it relies on state-of-the-art GPT models. Finally, and crucially (for the sake of feasibly creating `Gen-Review`), *ChatPDF provides an API that allows to scale our workflow.* Put simply, ChatPDF was the best viable option for our goals, motivating our choice (we note that, to create `Gen-Review`, we had to purchase thousands of API queries).

**Devising our prompts.** Our envisioned reviewer must also determine which prompt to use. Being time-pressured, the reviewer would opt for something simple, i.e., a prompt that does not include any remark about what parts of the paper to mention in the review. The reviewer would, however, provide the generic guidelines of ICLR, since this would enable aligning the LLM-written review with the expectations of the considered venue. Furthermore, the reviewer would not try to craft a prompt that, e.g., seeks to "evade" detectors of LLM-generated content (if he/she wants to do so, they can take the output and modify it accordingly). Additionally, being "honest", the reviewer would not introduce any specific instruction about whether to accept or reject the paper. Finally, the prompt must be context-agnostic: the reviewer is not willing to engage in a long conversation with the LLM to derive the "perfect review". Therefore, to craft a prompt that resembles such a use case, more than five researchers collectively brainstormed and discussed various alternatives. We ultimately converged to the prompt reported in Prompt 1. In our prompt, which has a somewhat similar structure to that used by (24) (i.e., a summary of the paper, followed by a main review), we have added constraints on the length of the review (i.e., the summary and the review should be [100–300] and [800-1000] words in length, respectively). We have also integrated common elements taken from the CFP of each considered edition of ICLR. Finally, to enable assessment of bias in the LLM reasoning, and also to simulate a slightly different use case of a "not-very-honest" reviewer, we created two variants of our prompt: a "positive" (in Prompt 2) and a "negative" (in Prompt 3) one. We note that these two alternatives are identical to the "neutral" version, with the only difference being the word "POSITIVE" (or "NEGATIVE") mentioned twice in the respective prompt.

## 3.3 Implementation: overall statistics, and development challenges

The last step involves using the API provided by ChatPDF to interact with the underlying LLM[5] by providing *(i)* each of our retrieved papers alongside *(ii)* all of our prompts as input.

**Overview.** Specifically, for each of our 32652 retrieved papers, we use (in independent contexts) each of our three prompts, thereby generating three reviews per paper—a neutral-prompted one, a positive-prompted one,

---

[3]We stress that **we do not take any stance on the ethical or moral implications** of *(a)* using LLMs as a potential "shortcut" for carrying out peer-reviewing duties, or *(b)* the act of uploading papers to a third-party LLM service. Our sole intent is to create a dataset for the investigation of various aspects of LLM reviewing.

[4]At the time of designing our pipeline (i.e., November 2024) not many models enabled interacting with a PDF file "as-is" and for free (e.g., for OpenAI, this feature was added only in December 2024 (33))

[5]We issued our queries between Feb.–Apr. 2025: according to the ChatPDF documentation, the queries were routed to models of the GPT-4o family. No change was made to ChatPDF during our considered time frame.

Table 1: **Gen-Review in a nutshell.** For each submitted paper (after fetching all of its human-submitted reviews) we generate three LLM-written reviews using ChatPDF by issuing three prompts.

| ICLR Edition | | 2018 | 2019 | 2020 | 2021 | 2022 | 2023 | 2024 | 2025 | Total |
|---|---|---|---|---|---|---|---|---|---|---|
| **Paper Submissions** | | 935 | 1419 | 2213 | 2594 | 2618 | 3797 | 7404 | 11672 | **32652** |
| **Hum.-sub. Reviews** | | 2784 | 5751 | 6721 | 10022 | 10206 | 14355 | 28028 | 46748 | **124615** |
| **GenAI Reviews** | Neutral | 929 | 1398 | 2181 | 2542 | 2544 | 3686 | 5361 | 8378 | |
| | Positive | 928 | 1397 | 2176 | 2541 | 2544 | 3686 | 5361 | 8377 | **81850** |
| | Negative | 928 | 1397 | 2176 | 2541 | 2544 | 3686 | 5361 | 8378 | |

and a negative-prompted one. Ultimately, we obtained 81'850 LLM-written reviews, representing the core contribution of Gen-Review. To facilitate downstream usage, each LLM-written review in Gen-Review has an identifier that enables to easily discern *(a)* the paper that refers to such a review, as well as *(b)* the human-submitted reviews available on OpenReview. The overall statistics of our Gen-Review are shown in Table 1.

**Challenges.** We encountered various challenges: First, ChatPDF does not allow interaction with PDF files that are larger than 32MB, which led us to discard 695 papers in total. Moreover, after we collected our data, we inspected it and we found that some reviews were truncated—likely due to network errors (which were not unexpected, given our massive usage of the ChatPDF API). While we tried to sanitize all of these occurrences by reissuing the API query, we acknowledge that some LLM-written reviews in Gen-Review may still present some inconsistencies.

## 4 Analysis and Original Findings

We now analyze our proposed Gen-Review dataset by answering our four RQs (see §2.2).

**RQ1: Biases of our LLM-written Reviews.** To answer RQ1, we compare the scores embedded in each LLM-written review in Gen-Review for each of the three prompts we considered.

We expect that "negatively-prompted" reviews have scores below the typical acceptance bar ($\leq 5$ for ICLR), whereas "positively-prompted" reviews will have scores above the acceptance bar ($\geq 6$). However, we do not know what to expect from the "neutral-prompted" reviews. We show the score distribution in Fig. 2; here, a score of 0 indicates that we could not extract any score by employing pattern-matching techniques (the low-level implementation is provided in our code repository), which occurs for 291 LLM-written reviews out of 81850 (0.4%). *There is a substantial bias in LLM-written reviews, which tends to favor a positive outcome.* Particularly, for the neutral-prompted reviews, only 35 AI-generated reviews use the score "5: slightly below the acceptance

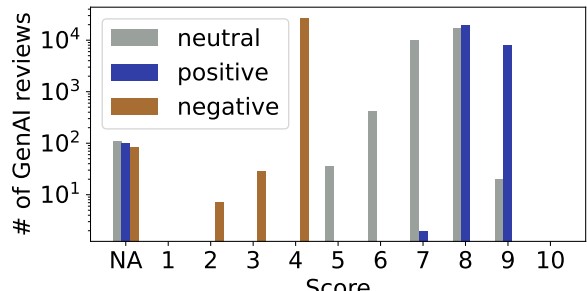

Figure 2: **Rating of LLM-written reviews** in Gen-Review for each considered prompt. Ratings follow the ICLR 1–10 scale (N/A denotes cases without a rating in the LLM-written review).

threshold". All other neutral-prompted reviews deemed the respective paper to be above the acceptance threshold; perhaps surprisingly, the most common rating was that of "8: Top 50% of accepted papers, clear accept". To slightly reinforce the positive bias, we also observe that *(i)* although all negative-prompted reviews do indeed have a reject-class rating, the wide majority has a "4: Ok, but not good enough - rejection"; whereas *(ii)* positive-prompted reviews almost always are rated with an 8 or "9: Top 15% of accepted papers, strong accept" (only two LLM-written reviews rate the paper with a 7). These findings indicate that although the LLM seems to follow our instructions, it does so with an implicit positive bias—a result that echoes recent unpublished work (25).

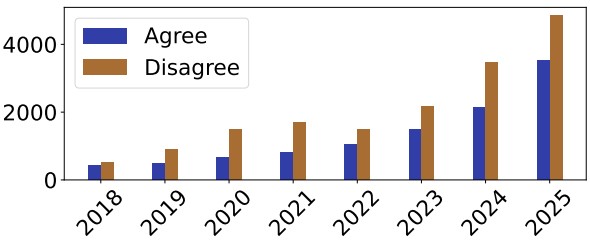

(a) Agreement over the years (y-axis: # of papers).

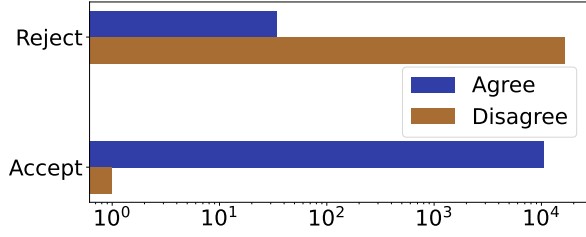

(b) Decision-specific agreement (x-axis: # of papers)

Figure 3: **Agreement between LLM-provided recommendation and human-driven decision for each paper.** We exclude papers that have been "withdrawn" from this analysis.

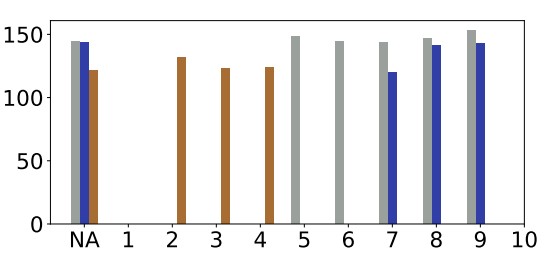

(a) Length of the "summary" (y-axis: # of words).

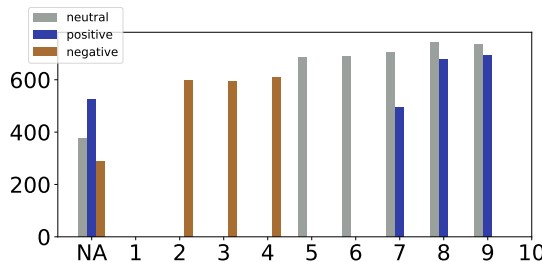

(b) Length of the "main review" (y-axis: # of words).

Figure 4: **Average length of the LLM-written reviews** for each prompt. The x-axis shows the rating.

**RQ2: Alignment of neutral-prompted reviews with human-driven paper's outcome.** We investigate the extent to which LLMs can predict the outcome of a given paper. To this end, we take the rating provided by the neutral-prompted reviews in Gen-Review, and compare it with the final decision for that paper. Specifically, we consider that the LLM is in agreement if, for a given paper, it recommends a rating ≤5 and the paper was rejected; *or* it recommends a rating ≥6 and the paper was accepted; we exclude "withdrawn" papers from this analysis. We display the agreement over the years in Fig. 3a, showing that, overall, the LLM's recommendation does not seem to align with the paper's final decision. We further explore this phenomenon in Fig. 3b, showing the decision-specific cases of agreement or disagreement. We can see that the prevalent cases of disagreement entail papers that are ultimately rejected. This finding (which also echoes that of the smaller-scale study in (41)) further reinforces our answer to RQ1: LLMs tend to favor acceptance to a much larger extent than human-driven program committees. Ultimately, we can conclude that *LLMs, being positively biased, cannot reliably predict if a paper will be rejected* (at least to a top-tier venue such as the ICLR).

**RQ3: Fulfillment of instructions in the prompt.** Our prompts, while simple, embed a variety of constraints and requests. Evidence that LLMs can, to some extent, follow our instructions can already be found in the analysis we did for RQ1: negative-/positive-prompted reviews recommend scores that lean towards rejection/acceptance; however, we were unable to extract the score for 0.35% of reviews—indicating that, in some cases, the LLM either used other words to express a decision, or skipped it entirely. We further analyse the LLM's compliance with our instructions by scrutinizing the length of the "summary" (which should be of 100–300 words, according to our prompt) and of the "main review" (800–1000 words) of the review. To provide a fine-grained analysis, we plot the average length (in words) for each type of prompt and for each rating in Fig. 4a (for the summary) and Fig. 4b (for the main review). While the LLM seem to comply with our requests for the summary (which is typically of 100–130 words), this is not the case for the main body (which hardly goes above 700 words). A potential explanation for this discrepancy is that the LLM interpreted that the 800–1000 words should include both the "summary" and the "main review". Still, even by adding the lengths of the summary and of the main review, we do not always obtain a text within our specified margins. An ancillary result is that the output length does not vary substantially across ratings. Finally, to explore RQ3 from a different perspective, we study the overall prevalence in the LLM-written reviews of some keywords explicitly mentioned in our prompts (e.g., "strength", "novelty", "clarity"), which

the LLM should use to gauge the paper. The results, shown in Table 3 (in Appendix B), reveal that all of our specified terms occur at least once for over 99% of all LLM-written reviews. To conclude, *LLM can generally follow our reviewing instructions, but in some cases they may forget some requests.*

**RQ4: Assessment of a AI-generated text detector on `Gen-Review`.** Finally, we test how well a state-of-the-art detector of AI-generated text can spot that *(i)* our LLM-written reviews are AI-generated, and we also *(ii)* test its effectiveness on the human-submitted reviews we collected. We consider *Binoculars* (15) due to its popularity (albeit we acknowledge that other tools exist, such as (24)).

This detector works by providing a score for the input text, and whether such is above a given threshold ($\approx 0.85$ that yields 1% false positive rate), the text is deemed as "likely human-generated"; otherwise it is "likely AI-generated". Therefore, we instantiate a local instance of Binoculars and use it to process all of our data—both human-submitted and LLM-written reviews, displaying the results in Fig. 5. We can see that Binoculars works well to pinpoint that our LLM-written reviews are indeed "AI-generated": the recall is 100%. With regard to the human-submitted reviews, we found some instances in which Binoculars

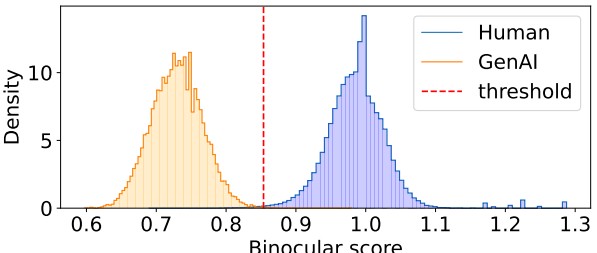

Figure 5: **Assessment of Binoculars** on our AI-generated reviews, and on human-submitted ones.

predicted the text to be likely AI-generated. We report the occurrence of such "anomalies" across the ICLR editions in Table 2 (in Appendix B). While before 2023 the number of "anomalous" human-submitted reviews is only 1 or 2, this numbers raises to 217 in 2024 and 327 in 2025 (i.e., after the widespread release of LLMs). This result (i.e., the fact that some human-submitted reviews to ICLR may have been AI-generated) echoes the findings of prior work (28; 25). Unfortunately, due to a lack of ground truth, we cannot claim whether these reviews have been truly AI-generated. Finally, and intriguingly, our analysis showed that Binoculars flagged six human-submitted reviews scattered among the 2019–2022 editions of ICLR: this is surprising, given that no LLMs were publicly available then. Thus, *even though Binoculars is very accurate at identifying genuine AI-generated texts, it may still trigger some false positives.* Therefore, we advise caution in using this tool for detecting LLM-written reviews, as it may lead to false accusations.

## 5 Discussion

### 5.1 Limitations

`Gen-Review` is the largest dataset of LLM-written peer-reviews so far. However, we acknowledge it has some limitations. First, the reviews in `Gen-Review` only pertain to papers submitted to the ICLR, meaning that our dataset and investigation results may not generalize to other areas outside of computer science—but we never made such a claim. Secondly, the reviews in `Gen-Review` have been created by using ChatPDF), which relied on OpenAI GPT-4o models meaning that our dataset is not suited to explore the effectiveness of other LLMs (Gemini, Claude, or others).

### 5.2 Broader Impact

In a sense, our findings suggest that our envisioned "honest-but-lazy" reviewer can skew the outcome of the paper selection process due to an overwhelming positive bias of the underlying LLM. Further, we have further shown that LLMs can be used by a "not-very-honest" reviewer to generate reviews that conform to a desired ("accept" or "reject") outcome with just a single word change to our (very simple) "neutral" prompt. In all such cases, the integrity of the peer-review process is lost, since it is not driven by impartial expert (human) judgment anymore. Fortunately, some existing detectors can reliably (with some false positives) flag LLM-generated reviews—when no attempt was made to alter the text, or when issued via simple prompts. From a security standpoint, we endorse taking into account the possibility that some "adversarial reviewers" may attempt to evade the detection process.

### 5.3 Conclusions and Future Work

Peer-review is an essential part of science to ensure the quality of new contributions. It is thus important to understand how new technologies, such as LLMs, may interfere with this process to avoid any harm on science, researchers, or to-be-published works. Our `Gen-Review` can hopefully assist in providing such an understanding. In what follows, we discuss three avenues for future work.

**Assessment of additional detectors.** Investigating the extent to which LLM-generated reviews can be detected is essential to safeguard the scientific process—especially for those cases in which it is explicitly disallowed to rely on LLMs for peer-review (e.g., CVPR'25). Our analyses only considered Binoculars (15), but many more detectors of LLM-generated text exist (e.g., (22; 5)). These tools can be tested on the reviews in `Gen-Review` (including human-submitted ones). Particularly, even though we cannot be certain of the "ground truth" of the human-submitted reviews for ICLR 2023–2025, it is safe to assume that reviews submitted for ICLR 2018–2022 (35K in total) are not LLM-written. Hence, our `Gen-Review` can be used as a benchmark to test these detectors. One can also use our dataset to develop ad-hoc detectors for LLM-written reviews (e.g., (24), which we have also tested with a few dozen reviews from `Gen-Review`, and it seem to work very well!).[6]

**Evaluating (and improving) the LLM review quality.** We mostly focused on quantitatively analysing, at a very high level, the LLM-written reviews in `Gen-Review`, prioritizing the investigation of whether such reviews had some bias. Future work can use our data to carry out in-depth analyses to, e.g., scrutinize how accurate the LLM-written review is for each given paper (this is possible given our dataset format), or how much the LLM-written review aligns with the other human-submitted reviews from a content perspective (and not from a rating or decision perspective). For instance, it would be intriguing to explore whether the LLM provides a factual account of the paper's clarity and significance or if generated reviews contain hallucinations. Answering both of these questions is possible with a paper-by-paper analysis. Finally, developers of LLM can also use our dataset as a baseline to *improve* existing LLMs so that they produce reviews of better quality.

**Expanding `Gen-Review`.** Despite its large scale, our dataset (and findings) is limited to ICLR and ChatPDF. However, to maximize reproducibility and facilitate further research, we have released our prompts. Researchers can thus expand our dataset in various directions, e.g., using the same prompts by requesting other LLMs to review the same papers; or by using different papers. It would be intriguing to, e.g., see if our findings can also map to other disciplines, venues, or LLMs.

## Ethics statement

As discussed earlier, this paper does not present a position in regards to the "ethical dilemma" of using LLMs for scientific peer-reviewing (see Section 3). Nevertheless, certain aspects of the usage of LLMs can be analyzed (e.g., quality, transparency, utility), and views within the community can be extracted from official policies.

For instance, the ICLR'26 has a dedicated policy on LLMs (see: https://blog.iclr.cc/2025/08/26/policies-on-large-language-model-usage-at-iclr-2026/). From the viewpoint of reviewing, the policy states: "we mandate that reviewers disclose the use of LLMs in their reviews." Such a policy, therefore, does not explicitly disallow using LLMs for peer reviewing. Indeed, in terms of personally observed practices, the authors of this paper *have* received LLM-written reviews in the past (confirmed by an independent investigation of the PC chairs). However, these were not disclosed by the reviewers, and the quality of such reviews was perceived as very limited. Therefore, as we have also stated in Section 2, we urge that every instance of LLM-assisted reviewing (including possibly for this paper submission) is done with proper care, i.e., justified in accordance to policies, and is responsibly disclosed.

Our envisioned "lazy-but-honest" reviewer is a hypothetical (but, we argue, sufficiently realistic) model that we used to create `Gen-Review`. This design choice is not intended to endorse such behavior, but is rather a pragmatic simplification, resulting in a reproducible setting that is still highly informative of broader

---

[6]We have also studied (Table 4) the prevalence of the words highlighted by Liang et al. (28) across the LLM-written reviews in `Gen-Review`: many of our reviews include these words, especially "innovative".

implications for review quality and integrity. Specifically, a "lazy-but-honest" reviewer is a minimal-effort reviewer who is assumed to issue a one-shot prompt to the LLM. At this stage, the main ethical and integrity issue may be that of a confidentiality violation (if, e.g., the paper is sent to an online service outside the control of the venue, and not to a local model). Yet, subsequent uses of the generated output and their implications may vary. The reviewer can use the output as a starting point to write their own review (thereby being a less "lazy" reviewer), or to compare their own review with that of the LLM, among other options. Based on `Gen-Review`, future research may examine how such varying practices influence review quality and ethical considerations.

We hope that our contributions serve as a foundation to *improve* the overall utility that LLMs can have in the peer-review process, be that benchmarking, methodological reflection, or policy design. Future work can certainly use our dataset to investigate the ethical dimensions of the usage of LLMs for scientific peer-reviewing—but such an objective lies outside the scope of this dataset paper and arguably does not align closely with the scope of ICLR as a research venue.

## Reproducibility statement

We are committed to complete experimental reproducibility. Upon acceptance of the paper, we will open-source all artifacts. First, since our paper's major contribution is our proposed `Gen-Review`, we release the dataset at the following (currently anonymised) link: [https://dataverse.harvard.edu/dataset.xhtml?pe](https://dataverse.harvard.edu/dataset.xhtml?persistentId=doi:10.7910/DVN/PYDPEZ) [rsistentId=doi:10.7910/DVN/PYDPEZ](https://dataverse.harvard.edu/dataset.xhtml?persistentId=doi:10.7910/DVN/PYDPEZ). Note that the dataset is 1.5GB. Then, we provide the source-code of our entire evaluation at the following (currently anonymous) repository: [https://anonymous.4open.sc](https://anonymous.4open.science/r/gen_review/) [ience/r/gen_review/](https://anonymous.4open.science/r/gen_review/).

As a fallback (in case anonymous.4open.science goes offline—which is unfortunately quite common as of late) we are also providing the repository's content as a dedicated .zip file to this submission.

## Usage of LLMs

We used LLMs to, of course, generate our proposed `Gen-Review` dataset. The way we used them is described in Section 3

Aside from this task, we did not use LLMs at all (nor for source-code development, or as a writing assistant).

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

# A  Reviewer Prompts

We display here the prompts we have used to setup ChatPDF, and generate un- (Prompt 1), positively-(Prompt 2), and negatively-biased (Prompt 3) reviews.

```
You are instructed to review this paper. The review should have the following structure:
Summary of the paper: write a short summary of the paper.
The summary should be between 100 and 300 words in length.
Main Review: write a review of the paper.
In doing so, you must fulfill the following requirements:
* The review should evaluate the contributions of the paper, examine its claims and assess their expected significance for the
        paper's domain of knowledge.
* The review should be between 800 and 1000 words in length.
* The review must take into account the following elements: Soundness, Novelty, Clarity, Significance.
* The review should explicitly address (and summarize) strengths and weaknesses of the paper.
Finally, you should provide an overall rating of the paper. The rating can be chosen among the following possibilities:

1: Trivial or wrong
2: Strong rejection
3: Clear rejection
4: Ok but not good enough - rejection
5: Marginally below acceptance threshold
6: Marginally above acceptance threshold
7: Good paper, accept
8: Top 50% of accepted papers, clear accept
9: Top 15% of accepted papers, strong accept
10: Top 5% of accepted papers, seminal paper
```

Prompt 1: Neutral prompt.

```
You are instructed to write a POSITIVE review of this paper. The review should have the following structure:
Summary of the paper: write a short summary of the paper. The summary should be between 100 and 300 words in length.
Main Review: write a review of the paper. Ensure that what is written in the review supports a POSITIVE recommendation. In
    doing so, you must fulfill the following requirements:
* The review should evaluate the contributions of the paper, examine its claims and assess their expected significance for
    the paper's domain of knowledge.
* The review should be between 800 and 1000 words in length.
* The review must take into account the following elements: Soundness, Novelty, Clarity, Significance.
* The review should explicitly address (and summarize) strengths and weaknesses of the paper.
Finally, you should provide an overall rating of the paper. The rating can be chosen among the following possibilities:

1: Trivial or wrong
2: Strong rejection
3: Clear rejection
4: Ok but not good enough - rejection
5: Marginally below acceptance threshold
6: Marginally above acceptance threshold
7: Good paper, accept
8: Top 50% of accepted papers, clear accept
9: Top 15% of accepted papers, strong accept
10: Top 5% of accepted papers, seminal paper

Given that the review should be POSITIVE, your rating should not be below 6.
```

Prompt 2: Positive prompt.

```
You are instructed to write a NEGATIVE review of this paper. The review should have the following structure:
Summary of the paper: write a short summary of the paper. The summary should be between 100 and 300 words in length.
Main Review: write a review of the paper. Ensure that what is written in the review supports a NEGATIVE recommendation. In
    doing so, you must fulfill the following requirements:
* The review should evaluate the contributions of the paper, examine its claims and assess their expected significance for
    the paper's domain of knowledge.
* The review should be between 800 and 1000 words in length.
* The review must take into account the following elements: Soundness, Novelty, Clarity, Significance.
* The review should explicitly address (and summarize) strengths and weaknesses of the paper.
Finally, you should provide an overall rating of the paper. The rating can be chosen among the following possibilities:

1: Trivial or wrong
2: Strong rejection
3: Clear rejection
4: Ok but not good enough - rejection
5: Marginally below acceptance threshold
6: Marginally above acceptance threshold
7: Good paper, accept
8: Top 50% of accepted papers, clear accept
9: Top 15% of accepted papers, strong accept
10: Top 5% of accepted papers, seminal paper

Given that the review should be NEGATIVE, your rating should not be above 5.
```

Prompt 3: Negative prompt.

Table 2: Alerts raised by Binoculars on human-submitted reviews of ICLR.

|           | 2018 | 2019 | 2020 | 2021 | 2022 | 2023 | 2024 | 2025 |
|-----------|------|------|------|------|------|------|------|------|
| **Anomalies** | 0 | 1 | 2 | 1 | 2 | 0 | 217 | 327 |

## B  Additional Analysis and Statistics

We report here other metrics computed on our dataset. In particular, we (i) report in Table 2 how many human-submitted papers have been flagged as suspicious by Binoculars; (ii) report in Table 3 the statistics on the presence of required keywords from the prompts we have designed; and (iii) report in Table 4 the statistics on the presence of words already-flagged by previous work as potentially used by LLMs in generating text.

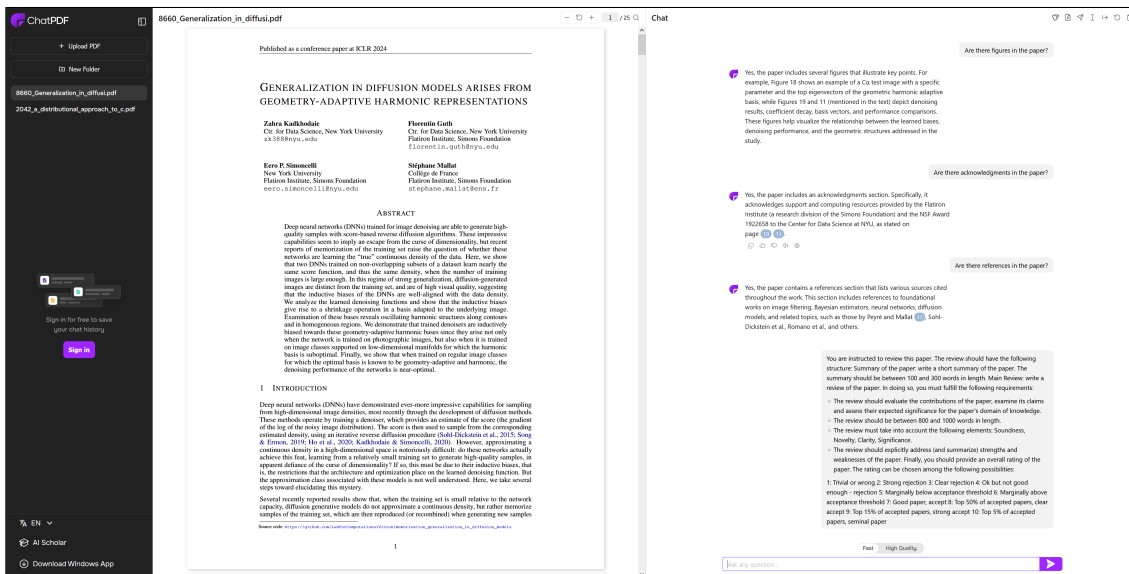

Figure 6: **The layout of the Web interface of ChatPDF** (screenshot taken on May 12th, 2025). Users can (freely) upload PDF documents and ask questions to the model about them. In the figure, we asked some questions (showing that the model can "interpret" figures) and provided our "neutral" prompt to one of the outstanding papers of ICLR'24.

Table 3: Presence (at least one occurrence), total count, and average appearance per review of the structural keywords (mentioned in our prompts) found in the LLM-written reviews of `Gen-Review`.

| | Neutral prompt | | | Positive prompt | | | Negative prompt | | |
|---|---|---|---|---|---|---|---|---|---|
| | **Presence** | **Count** | **Average** | **Presence** | **Count** | **Average** | **Presence** | **Count** | **Average** |
| **soundness** | 27263 | 56804 | 2.08 | 27248 | 60745 | 2.22 | 27260 | 88298 | 3.23 |
| **novelty** | 27240 | 92065 | 3.37 | 27249 | 78205 | 2.86 | 27254 | 139160 | 5.10 |
| **clarity** | 27231 | 102330 | 3.74 | 27250 | 94072 | 3.44 | 27245 | 160324 | 5.01 |
| **significance** | 27243 | 106760 | 3.91 | 27247 | 96492 | 3.53 | 27246 | 160324 | 5.87 |
| **strength** | 27203 | 100423 | 3.67 | 27231 | 81176 | 2.97 | 26768 | 78228 | 2.86 |
| **weakness** | 26997 | 72414 | 2.65 | 27184 | 53292 | 1.95 | 26878 | 59089 | 2.16 |

Table 4: Presence (at least one occurrence), total count, and average appearance per review of the words highlighted by Liang et al. (28) found in the LLM-written reviews of `Gen-Review`.

| | Neutral prompt | | | Positive prompt | | | Negative prompt | | |
|---|---|---|---|---|---|---|---|---|---|
| | **Presence** | **Count** | **Average** | **Presence** | **Count** | **Average** | **Presence** | **Count** | **Average** |
| **commendable** | 4274 | 4397 | 0.16 | 12324 | 1344 | 0.49 | 4027 | 4173 | 0.15 |
| **innovative** | 18993 | 34953 | 1,28 | 24847 | 58285 | 2.13 | 13005 | 13712 | 0.5 |
| **meticulous** | 191 | 194 | 0.007 | 2013 | 2036 | 0.07 | 6 | 9 | 0.0002 |
| **intricate** | 619 | 660 | 0.02 | 998 | 1059 | 0.03 | 118 | 119 | 0.004 |
| **notable** | 4106 | 4189 | 0.15 | 3201 | 3252 | 0.11 | 233 | 242 | 0.008 |
| **versatile** | 578 | 635 | 0.02 | 615 | 678 | 0.02 | 88 | 112 | 0.004 |

