# OpenReview forum: "GenReview: : A Large-scale Dataset of AI-Generated (and Human-written) Peer Reviews"
_TMLR — Rejected by TMLR_

### Review · Reviewer_dhJn · 2025-12-27

**Summary Of Contributions:**

The authors present "Gen-Review," a dataset containing approximately 81,000 AI-generated peer reviews and their corresponding human-written reviews, based on submissions to ICLR from 2018 to 2025. The authors generated these reviews using the ChatPDF API (based on GPT-4o) under three prompt settings: neutral, positive, and negative. The paper analyzes this dataset to explore research questions regarding AI bias, the alignment between AI scores and human acceptance decisions, instruction following, and the detectability of AI text using the "Binoculars" tool. The authors find that the generated reviews exhibit a strong positive bias and that detection tools perform well on this specific dataset.

**Additional Comments:**

The paper addresses a timely problem, and the scale of data collection (81k reviews) is impressive. However, quantity does not substitute for quality or diversity. A dataset intended to serve as a benchmark for the community must be representative of the problem space. By restricting the data to a single commercial tool (ChatPDF), the authors have created a dataset that detects "ChatPDF outputs" rather than "AI-generated reviews." This distinction limits the scientific contribution of the work below the bar for TMLR.

**Audience:**

Yes

**Audience Explanation:**

The topic of Large Language Models (LLMs) in peer review is currently of significant interest to the machine learning community. Understanding the prevalence, characteristics, and detectability of AI-generated reviews is crucial for maintaining the integrity of the scientific process. However, the interest depends heavily on the robustness and diversity of the data, which is currently lacking.

**Broader Impact Concerns:**

The authors briefly touch upon the risk of "adversarial reviewers" using this data to evade detection. However, a more pressing concern is the copyright and privacy implication of the methodology. The authors processed thousands of papers through a commercial third-party tool (ChatPDF). Even though ICLR papers are open access, mass-processing them through a commercial API that may or may not retain data for training raises ethical questions regarding the respect for the original authors' data. The authors should explicitly discuss the data retention policies of the tools they used.

**Claims And Evidence:**

No

**Claims Explanation:**

While the authors have successfully collected a large volume of data, the evidence provided does not sufficiently support the broader claims regarding the utility of this dataset for understanding "LLM-generated peer reviews" in a general sense.

1.
**Lack of Model Diversity:** The entire dataset is generated using a single commercial wrapper (ChatPDF). The authors acknowledge this limitation , but it is fatal for a dataset paper claiming to enable "broad investigations". Findings derived from this dataset (e.g., detectability, bias patterns) may be specific to the system prompt or formatting of ChatPDF and may not generalize to raw GPT-4o, Claude 3.5, Llama 3, or other models likely to be used by actual reviewers.


2.
**Limited Prompt Engineering:** The "lazy reviewer" assumption is interesting, but the prompts used are relatively simplistic. The study does not account for reviewers who might use more sophisticated prompting strategies (e.g., few-shot prompting with examples of high-quality reviews) to evade detection. Therefore, the claim that this dataset serves as a robust benchmark for detection  is weak, as it only represents the "lowest hanging fruit" of AI-generated text.


3.
**Detectability Claims:** The authors claim Binoculars has 100% recall on their dataset. This near-perfect performance is suspicious and likely an artifact of the specific, repetitive structure generated by ChatPDF (which likely has its own system prompt), rather than an indication that AI reviews are easily solvable. Without cross-model validation, the evidence is unconvincing.

**Requested Changes:**

To secure a recommendation for acceptance, the authors would need to significantly expand the scope of the dataset and the analysis. Given the scale of these changes, they likely constitute a new submission rather than a revision.

**Critical Adjustments:**

1. **Model Expansion:** The dataset must include reviews generated by a diverse set of state-of-the-art LLMs (e.g., Claude 3 Opus/Sonnet, Gemini 1.5 Pro, Llama 3, raw GPT-4o via API) rather than relying solely on the ChatPDF wrapper. This is necessary to determine if the "positive bias"  and detection results are model-agnostic or artifacts of one specific tool.


2. **Prompt Diversity:** Include a wider variety of prompts. The current "neutral/positive/negative" split is useful, but the dataset needs "sophisticated" prompts that attempt to mimic human style or use few-shot prompting. This would make the dataset a valid benchmark for adversarial detection.


3. **Comparative Analysis:** Compare the quality of the generated reviews against the human reviews using automated metrics (beyond simple word counts) or a small-scale human annotation study. Currently, the analysis focuses heavily on metadata (scores, length)  rather than the semantic content of the reviews.


4. **Legal/Ethical Clarification:** Please clarify the terms of service regarding uploading 32,652 PDFs to a third-party commercial service (ChatPDF). While ICLR papers are public, uploading them en masse to a commercial entity for processing raises data usage and copyright questions that should be addressed in the paper.

---

> ### Author Response · Authors · 2026-02-12
>
> Thanks for the review. We are pleased the reviewer observed that “the scale of data collection (81k reviews) is impressive”.
>
> Let us respond to some critical remarks:
> * “_Legal/Ethical Clarification_”: we observe that, in our paper, we explicitly wrote that our research method is compliant with OpenReview’s terms of use (see bottom of page-5). All information we used for our research is publicly available and provided with no copyright and, importantly, “submissions to ICLR cannot be deleted or modified” (from https://iclr.cc/Conferences/2026/AuthorGuide). Therefore, there is no “privacy” implication. Furthermore, ChatPDF allows users of its API to delete the uploaded papers (see https://www.chatpdf.com/docs/api/backend).
> * “_Lack of Model Diversity_”: with respect, we wrote “GenReview enables a broad range of investigations” (we wrote this only in the Abstract). We still believe this is the case, given that our dataset encompasses papers (and reviews) submitted over 8 years of editions of a venue that receives submissions from a variety of research domains, and we also used multiple prompts. Moreover, we justified our choice for using ChatPDF, as it was a popular tool for interacting with PDFs. Finally, please note that models within the GPT family are the most widespread ones.
> * “_Prompt diversity_”: our assumption is that of a “lazy” reviewer, and investigations of “sophisticated” prompting strategies are outside our scope. Specifically, evaluations of a, virtually infinite, search space such as prompts for a review is beyond the scope of a sole research paper, let alone ours which is focusing on a dataset. Given the lack of datasets with a scope similar to GenReview, contributing with one such dataset is important (as also remarked by the two other reviewers).
>
> Nevertheless, we are committed to carry out the following **changes**, which are meant to address the outstanding remarks and cover the changes:
> * “Model Expansions”: we can expand our dataset and results analysis by considering LLMs of different families than GPT ones (GPT models are used by ChatPDF).
> * “Detectability”: we can expand our assessment of detectability of LLM-written (and human-submitted) reviews by including a commercial detector, Pangram.
> * "Ethical/Legal": we can certainly expand on this by providing more details on the legitimacy of our approach.
>
> From a feasibility viewpoint (in terms of financial cost and human effort), it is unfeasible to replicate our entire dataset of 81k reviews by using additional LLMs. But doing so for a subset of it, so that we can gauge if the results hold also for other LLMs, is feasible and we can certainly (and happily) undertake such an effort. Under these terms, these changes can be implemented in a revision, and do not require a resubmission.
>
> Thanks again—also for pointing out that TMLR's audience could be interested in our findings.

---

### Review · Reviewer_x68N · 2026-01-10

**Summary Of Contributions:**

This paper proposes Gen-Review, a large-scale dataset designed to facilitate research into the role of LLMs in scientific peer review. The authors collected metadata, PDF submissions, and more than 124,000 human-written reviews for all papers submitted to the ICLR between 2018 and 2025. Using this data, then the authors generated 81,850 synthetic reviews. To simulate different reviewer behaviors, they employed three distinct prompts: neutral, positive, and negative. Finally, the paper shows the analysis of this dataset and provides several key findings: 1) "Neutral" LLM reviewers exhibit a strong bias toward acceptance, often rating papers significantly higher than human reviewers. 2) While LLM ratings align with accepted papers, they fail to reliably predict rejections due to this positive bias.

**Strength**
1. Motivation is clear, and the dataset is useful. The authors created a large empirical dataset to study the LLM-generated results, which is important to the research community.
2. The authors provide the open-sourcing the dataset and code, and have provided the specific prompts used.

**Weakness**
1. I have a concern about the quality metric. The analysis of the reviews focuses largely on word-based metadata (scores, word counts, keyword presence). The paper checks if the LLM used the word "novelty", but not whether the LLM correctly identified the novelty or lack datails. Encouraging the author to do some preliminary evaluation of content validity is expected to prove the dataset's utility for quality assessme.

2. The authors only note that the Binoculars detector flagged 6 reviews from 2019–2022 period as AI-generated. While this number is low, it casts some doubt on the detector's performance when applied to the 2024–2025 data.


3. The dataset quality may rely on "ChatPDF" and "GPT-4o" as a black box. While they argue this simulates a "lazy reviewer" who uses off-the-shelf tools, it introduces uncontrolled/biased variables.

**Audience:**

Yes

**Audience Explanation:**

Abusing LLM in peer-rewiew process is an emergency issue for the current research community. By quantifying phenomena that have largely been anecdotal, this paper grounds the debate in new empirical dataset.

**Broader Impact Concerns:**

The author provides the "Ethical Statement". I think it's ok, and No further ethical concerns for this paper.

**Claims And Evidence:**

Yes

**Claims Explanation:**

The claims made in the submission are almost supported by paper content. The authors are generally careful to the qualify their findings, acknowledging limitations like the lack of ground truth for human reviews and the restriction to a single LLM provider.

**Requested Changes:**

1. See in Point 1 of "Weakness".

2. For point 2, could the author provide some preliminary results by applying Binoculars detector to the recent data?

3. For point 3, could the author provides more detail step and anaysis for "ChatPDF"? For exanple, does ChatPDF chunk the PDF text and does it have a context window limit that might cause it to miss some part of original paper?

---

> ### Author Response · Authors · 2026-02-12
>
> Thank you for remarking that our “Motivation is clear, and the dataset is useful”! We are also pleased to read that you recognised that “Abusing LLM in peer-review process is an emergency issue for the current research community”.
>
> We acknowledge your constructive remarks. We believe that, with a minor revision, we can address them appropriately. Specifically, we are prepared to make the following changes:
> * **“Novelty”**: we can _qualitatively inspect_ a subset of the LLM-generated reviews to infer if there is any form of assessment on the “novelty” of the corresponding paper.
>   * While we cannot do so for the entirety of our dataset, we can provide a more informed conclusion on whether LLMs do follow instructions embedded in our reviewing prompts.
>   * We can apply some heuristics in choosing the considered reviews (E.g., some mentioning the word “novelty”, and some not doing so).
> * **Binoculars**: We did not fully understand the request about “applying Binoculars detector to the recent data”: in our paper, _we report the results of Binoculars for all of our dataset_ (see Table 2, in which we report the number of reviews flagged by Binoculars between 2018–2025). Could you clarify?
>   * What we can do, however, is manually checking if the human-submitted reviews flagged by Binoculars can—in our opinion—be deemed as being (likely) “LLM-written”.
>   * We can, moreover, check if these human-submitted reviews are also deemed as being (likely) LLM-written according to other detectors of LLM-generated content (e.g., we can use Pangram).
>
> **About ChatPDF**: please note that ChatPDF is a black box to us, because we are not affiliated to it. Therefore, any conclusion we would draw would just be a conjecture—which is why we refrain from analysing ChatPDF in detail. Nonetheless, we observe that, in our paper, we _do report some limitations/known-issues of ChatPDF_ that we had to deal with. See the paragraph “Challenges” on page-7 of our submitted paper.
>
> In summary, our proposed changes above would make our contributions stronger, and are humanly feasible to carry out in a minor revision. Thank you again for your constructive comments!

---

### Review · Reviewer_nUjf · 2026-01-31

**Summary Of Contributions:**

The paper introduces gen-review, a dataset of ~82000 LLM generated reviews using ICLR papers from 2018-2025. Authors use ChatPDF to parse the papers and to generate the reviews. They generate three reviews per paper (positive, negative, and neutral). The generated reviews are lined to human reviews from the respective conference cycles. In addition to the dataset, the authors investigate (1) Intrinsic bias in LLM generated reviews (2) alignment of neutral reviews with the paper outcome (3) LLM's ability to follow the instructions in the prompt and (4) How well LLM generated text identifiers can identify these generated reviews.

Main strengths of the paper:
- The dataset introduced is the largest publicly available dataset for peer reviews and including human reviews makes it even stronger
- The research questions are well motivated to be asked using such a dataset
- Authors provide access to the dataset, prompts, enabling reproducibility

Main weaknesses of the paper:
- The paper uses a single LLM, ChatPDF API calls, which relies on GPT-4o. It would be good to see the variation in data and the results of research questions using other LLMs.
- The current dataset is only restricted to ICLR, which narrows the scope quite a bit.
- There is very limited discussion around the use of such a dataset and the value added to the community

**Additional Comments:**

None

**Audience:**

Yes

**Audience Explanation:**

Yes, I think there would be interest in the community for the dataset.

**Claims And Evidence:**

Yes

**Claims Explanation:**

The main claims in the paper are the large dataset made available, which the authors do make publicly available. Other claims around the research questions can be improved by more robust experiment design using multiple LLMs, more reviews than just ICLR, and using statistical significance.
I feel that diving a bit deeper into why some of these biases occur would be helpful as well and also discussion around what this dataset shows, the overarching conclusion, that we are not there yet to use LLMs for peer review, and to replace human reviewers.

**Requested Changes:**

- If possible, please consider other LLMs, including open source offerings
- Consider expanding to other conferences, however, I do understand if it is out of scope for this paper
- Add deeper discussion around the results, not just what the results show, but why that can be
- Include discussion on where this data shows we are in respect to using LLMs for peer review, as that question comes up often

---

> ### Author Response · Authors · 2026-02-12
>
> Thanks for recognising that GenReview is “the largest publicly available dataset and including human reviews makes it even stronger” as the first strength of our contributions!
>
> We agree with your critical remarks. While we are unable to replicate our entire process via other LLMs (for economical reasons), we believe that the following changes are feasible to carry out:
>
> * Creation of additional examples by applying the same reviewing prompts (neutral, positive, negative) on a _subset_ of the papers considered in our dataset, and providing such inputs to **additional LLMs** of a different family than GPT-based ones.
>   * For instance, within reasonable computational/cost capacity, we can ask LLAMA or Gemini to provide a review by using our prompts on some of the papers in our dataset.
>   * We can then analyse the resulting reviews by following the same procedure used in the paper, and check if there are some similarities across LLMs
> * Addition of **statistical tests** to support our claims. For instance, we can statistically measure the alignment across the ratings provided by each reviewing prompt, as well as measure the confidence of our assessments. Such tests would undoubtedly “add depth to our results”.
> * Adding a discussion on “where we are in respect to using LLMs for peer review”. We can take the opportunity to discuss the pros-and-cons of the policies adopted by various venues.
>
> Respectfully, we believe that expanding to other conferences would not make our contributions significantly stronger: the ICLR is a broad conference and already encompasses many themes revolving around machine learning (e.g., applied cybersecurity, or financial forecasting), and it provides full information about its peer-review process. Other venues may not be as transparent (e.g., for papers rejected to NeurIPS, reviews are available only if authors opt-in, which leads to bias). If the reviewer can suggest another venue that fits our criteria, we are open to considering it.
>
> Nonetheless, we believe the changes above to be feasible to carry out in a “minor revision” and would substantially improve our contributions. Thanks!

---

### Author Response · Authors · 2026-02-12
**Meta Response to all Reviewers**

We thank the reviewers for their feedback and kind words of appreciation for our research.

We believe that, with a revision, we can enhance our paper so that it addresses the most explicit concerns emerging in the reviews.

Specifically these are the changes we are committed to make to our submission, inspired by the comments of all reviewers:

* **Evaluation of an additional detector of LLM-generated content.** We will evaluate Pangram (https://www.pangram.com/) on a subset of our dataset (encompassing both LLM-generated and human-submitted reviews) and compare its performance with Binoculars.
* **Evaluation of additional LLMs.** We will replicate a portion of our experiments (review generation and subsequent assessment) on reviews generated by LLMs of families different from those by OpenAI.
* **Further evaluation of our results** We will qualitatively analyse some reviews in GenReview (e.g., those flagged by AI-detectors for additional validation; and a subset of all other ones to inspect how the LLM analysed the “novelty” of the paper).

We respectfully maintain that _replicating our entire experiments_ on multiple LLMs and/or by considering semantically different prompts to be an unfeasible endeavor that is locked behind economical costs. Moreover, such evaluations—while undoubtedly informative—do not undermine our current contribution: even today, we are not aware of a dataset of LLM-written reviews that is comparable to our proposed GenReview.

If we are given a minor revision, we will gladly enhance our paper with the aforementioned changes (which will, moreover, also include additional changes which mentioned in the specific response to each reviewer). Ultimately, we are fully intent on improving our paper and we believe that following our revision plan substantially expands the quality of our contributions to the audience of TMLR.

Best regards,
The Authors

---

### Decision · Action_Editor_NUDA · 2026-04-13

**Recommendation:** Reject

**Additional Comments:**

### 1. Cross-Model Generalization

Currently, the findings rely heavily on ChatPDF/GPT-4o, which raises concerns that the observed "positivity bias" might be model-specific rather than a general LLM trait.

**Action required**: You do not need to regenerate the entire dataset. Instead, please sample a representative subset (e.g., at least 800 papers, 100 per ICLR year) and generate reviews using the same prompts with at least two non-GPT models (e.g., Claude 3.5, Gemini). Repeat your RQ1 and RQ4 analyses on this subset and use statistical tests (e.g., KS test) to demonstrate whether the positivity bias and detection rates hold across different model architectures.


### 2. Detection Robustness Probe (Adversarial Realism)
Testing AI detectors on raw LLM outputs naturally yields near-perfect detection rates, but this does not reflect real-world adversarial use where human reviewers might lightly edit the AI's output.

**Action required**: Conduct a small-scale probe (on a few hundred reviews) simulating "light human editing." This could include automated synonym replacement, paragraph reordering, or mixing human-written sentences with LLM paragraphs. Report how the detector's performance degrades under these conditions. An exhaustive analysis is not required, but acknowledging and quantifying this gap between your experimental setup and real-world behavior is essential.

### 3. Deeper Discussion of Results (Sections 4 & 5)
The current manuscript does an excellent job of describing what the LLMs do, but falls short of explaining why they do it.

**Action required**:  Discuss the underlying mechanisms causing the positivity bias (e.g., the impact of RLHF alignment, safety guardrails, or the lack of harsh critiques in the training data distribution). Furthermore, discuss the practical implications of your findings: where might LLMs actually complement human reviewers, where do they fail, and what specific policies should academic venues adopt regarding LLM usage based on your evidence?

### 4. Explicit Scope Justification
Focusing on a single venue (ICLR) and a single primary model is a potential point of criticism, but it is understandable given the massive scale of the dataset.

**Action required**: Add a dedicated paragraph explicitly justifying this design choice. Cite examples from other large-scale dataset papers to support your methodology. Frame the expansion to multi-model and multi-venue datasets as a community effort that will be enabled by your released prompts and pipeline.

**Audience:**

Yes

**Audience Explanation:**

LLM researchers

**Claims And Evidence:**

No

**Claims Explanation:**

The paper introduces Gen-Review, a dataset of AI-generated scientific peer reviews. Motivated by the increasing (and often undisclosed) use of Large Language Models (LLMs) in academic reviewing, the authors created this dataset to facilitate large-scale, data-driven research into the impact, quality, and detectability of LLM-assisted peer reviews.

This paper tackles a highly timely and critical issue: the infiltration of LLM-generated content in the academic peer review process. The proposed approach to establishing a "ground truth" dataset of AI-generated reviews and your findings regarding the "positivity bias" of LLMs are both fascinating and highly valuable to the community.

However, while the premise is strong, the reviewers have raised several valid concerns regarding the generalizability of the findings, the realism of the detection experiments, and the depth of the discussion. Consequently, the manuscript requires substantial revision before acceptance

**Resubmission Of Major Revision:**

The authors may consider submitting a major revision at a later time.